# Zero-Day Malware Detection and Effective Malware Analysis Using Shapley Ensemble Boosting and Bagging Approach

**DOI:** 10.3390/s22072798

**Published:** 2022-04-06

**Authors:** Rajesh Kumar, Geetha Subbiah

**Affiliations:** School of Computer Science and Engineering, Vellore Institute of Technology, Chennai Campus, Chennai 600127, Tamil Nadu, India; geetha.s@vit.ac.in

**Keywords:** machine learning, computer security, artificial intelligence, boosting, bagging, cyber security, zero-day vulnerability, zero-day malware detection, Shapley value

## Abstract

Software products from all vendors have vulnerabilities that can cause a security concern. Malware is used as a prime exploitation tool to exploit these vulnerabilities. Machine learning (ML) methods are efficient in detecting malware and are state-of-art. The effectiveness of ML models can be augmented by reducing false negatives and false positives. In this paper, the performance of bagging and boosting machine learning models is enhanced by reducing misclassification. Shapley values of features are a true representation of the amount of contribution of features and help detect top features for any prediction by the ML model. Shapley values are transformed to probability scale to correlate with a prediction value of ML model and to detect top features for any prediction by a trained ML model. The trend of top features derived from false negative and false positive predictions by a trained ML model can be used for making inductive rules. In this work, the best performing ML model in bagging and boosting is determined by the accuracy and confusion matrix on three malware datasets from three different periods. The best performing ML model is used to make effective inductive rules using waterfall plots based on the probability scale of features. This work helps improve cyber security scenarios by effective detection of false-negative zero-day malware.

## 1. Introduction

Malware are meant to exploit the vulnerability and exposure of various software product such as applications, Operating Systems (OS), drivers, etc. The popularity of OS and applications make them a hot target for malware attacks. The ten top vendors from the top 50 software vendors that have vulnerabilities in their various software products are listed in Table 1, and the ten top products from fifty top software products are listed in Table 2 from a common vulnerability and exposure website. The speed of the generation of malware is very high these days. AlienVault—Open Threat Exchange is a crowd-sourced computer-security platform. It shares more than 19 million potential threats daily among more than 80,000 participants from 140 countries. Malware authors have polymorphic and metamorphic engines for generating new malware at high speed. These malware are exploited to convert these threats into attacks. The polymorphic and metamorphic engines generate dissimilar malware variants for zero-day attacks. The polymorphic and metamorphic engines modify some parts of the source code of existing malware to produce a new malware variant. For instance, reassignment of the registers such as replacing [PUSH eax] with [PUSH ebx] and related changes for POP instructions replace code between registers by exchanging register names. The program behavior is the same as before. These methods change the hash values and signatures for the malware and it is not detectable by anti-virus software, which depends on signatures or hash values.

Pohl [1] stated that less than zero-day vulnerability means it is known to a restricted set of people such as hackers or dedicated security researchers and the vulnerability is not communicated to any software vendors or developers of the software. Egelman et al. [2] stated that zero-day vulnerability means that it is known to the software vendor or developer of software but the fix for the vulnerability has not been released. The hardware or software vendors cannot repair the vulnerability with a new patch before getting to know or discovering the zero-day vulnerability. The vendors and developers get zero-day to fix the vulnerability. In the meantime, the hackers who know how to exploit the vulnerabilities could attack using unknown malware.

Organizations dealing with security solutions find it difficult to give mitigation solutions for zero-day vulnerabilities. Hence, the attacks exploiting zero-day vulnerabilities are the toughest among various threats to defend against. The attackers can have the best breakthrough with a zero-day attack in an organization that has good security solutions in a place that is difficult to compromise. It is difficult to predict the type, location, and way of exploitation of zero-day attacks as zero-day exploitation is more complex and diverse. However, many times these vulnerabilities are exploited with variants of existing malware or with new malware. To make the systems secure, one has to have a good system to detect future malware. If a ML model is general enough to learn from currently known malware and detect future malware then the ML model is good for identifying zero-day attacks with malware as an exploit. One of the goals of this paper is to show the effectiveness and robustness of the machine learning model to predict future malware after being trained by previously known malware.

Many ML models to detect malware are based on static, dynamic, and hybrid analysis with various performances. But the ML models have no way of knowing top features, input by top features and other features to the prediction of malware, and effectively compare features for misclassification. We studied zero-day malware detection using ML models to determine top features, and input by top features to the prediction of zero-day malware. By knowing input by features, the reasons for misclassification by the ML model can be determined. These reasons can be used to overcome misclassification. This leads to improving the efficiency of the ML model. The amount of contributions of features for true positive and true negative can be found to find trends of top features and their input. The trends can be used to confirm the classification, making the ML model robust. The trend can also be used to differentiate misclassification from correct classification for unknown future malware (zero-day malware).

The prime motivation in this proposal is for various novel visualizations of top features for zero-day malware detection and input by each top feature in shap value and on a scale based on predicted probability value by the ML model. It takes four steps further in the realm of visualization of zero-day malware detection. The visualizations show: (1) Identification of top features; (2) Input by each top feature in shap value; (3) Input by each top feature on a predicted probability scale; (4) Trends in the top five to nine features and input by top features for zero-day samples that are predicted as False Positive, False Negative, True positive, and True Negative. This work uses two bagging ML models such as Random Forest and Extratree forest and two gradients boosting decision tree (GBDT) boosting ML models such as XGBoost and LightGBM for experiments, results, and comparison.

The advantages of visualization are its capabilities to present a huge amount of data interactively and intuitively. Visualization by bar plots and waterfall plots in shap value and probability scale display top features and their contribution. Visualization techniques used in this work can immensely aid the security analysts in a thorough analysis of the suspicious software and help them effectively deal with a large number of malware. Various agencies, such as anti-virus industries, threat detection agencies, and both open and invited malware data websites, maintain a large database of malware. Some of these are available and many may not be available to researchers. The contribution in this work may help in effective analysis to meet the complexity and scale of malware generation on a per-day basis.

This study has the following novelties:Use of two GBDT boosting and two bagging algorithm ML models for predicting the zero-day malware and benign software. The results are compared to determine the best performing ML model.Identification of change in top features for False Positive (FP), False Negative (FN), True Positive (TP), and True Negative (TN) categories.Determination of the contribution by each feature on a probability scale that has a direct correlation to the predicted probability by machine learning models. The top nine features are determined by the highest contribution on the probability scale.Determination of the contribution by each feature in Shapley value and identify the top nine features by the highest contribution.Compare the top features and contributions of features on a probability scale for each predicted category such as FP, FN, TP, and TN samples by the trained ML model. Identify the trends in top features and their input for FN and FP samples. The trends are made into inductive rules to improve the efficiency of ML models. The efficiency of the ML model is improved by inductive rules and not by feature selection or hyperparameter tuning.Use these trends for confirmation of TP and TN samples, making the ML model robust.

The ML model may detect malware but needs further improvement as false-negative misclassification can leave out detection of malware. The motivation here is to detect them with trends in top features based on Shapley value. The false-positive misclassification causes inconvenience to the user and is also planned to be reduced. This improves the efficiency of the ML model. The trends in top features can be used to confirm the correct classification, making the ML model robust.

This paper has a literature survey in Section 2, and methodology of zero-day malware detection and visualization in Section 3. The experimental setup, datasets, and results are described in Section 4 and the Conclusions in Section 5.

## 2. Literature Survey

Venkatraman et al. [3] used data visualization for zero-day malware, with an unbalanced dataset of 75,000 samples with 2/3 as malware and 1/3 as benign. They adopted two methods to visualize the zero-day malware. In the first one, the images of files were passed through a trained CNN model for feature extraction. The features were plotted as two-dimensional t-Distributed Stochastic Neighbor embedding for six families by K-means clustering. In the second method, they visualized eight different distance measures such as Cosine, Bray-Curtis, Canberra, Chebyshev, Manhattan, Euclidean, Hamming distance, and Correlation for n-gram of API calls from disassembled code of an unpacked sample. Their n value in the n-gram range was from 1 to 5. They used six categories of API call sequence, which are used by malware to access system services, to get input for the DLLs used, and to create or modify files. The eight different distance measures were used to draw a similarity matrix with Support Vector Method (SVM) and four different kernels, including RBF. Many malware authors use proprietary packing techniques that cannot be unpacked. This technique will not be able to unpack such malware and process them.

Yousefi-Azar et al. [4] generalized their model for zero-day, unseen malware by Malytics and Extreme Learning Machine (ETM). Malytics is inspired by natural language processing and the term frequency of malware and clean ware. The static features of a sample are extracted and classified by a deep learning model. To reduce the computational complexity of large feature space during backpropagation, they multiplied the random projection matrix with the term frequency of the sample. This is a linear process termed tf-simhasing. Malytics and ETM were used on the Drebin and Dexshare android dataset, which includes balanced malware and clean ware. These were also used with a windows executables dataset. They used windows malware from 2016 to build the model and windows malware from 2017 as zero-day malware to test against the model. They achieve an accuracy of 95.5%. We used the zero-day malware from the next month and got a much higher accuracy of 98.5%.

Kardan et al. [5] explained fitted learning concepts for ML models to make them aware of the limits of ML models. The authors visualized simple data to demonstrate the normal machine classification that is not able to distinguish outliers that should not be part of another class where a boundary is drawn for one class. A classic example of fooling in machine learning is the following example. To classify an apple with weight (0.2, 0.25) kilograms and watermelon with weight (5.2, 6.0) kilograms using SVM will have a boundary of 2.5 kg. It will classify anything of 1.5, 2.0 kg as apple and even 20 kg as watermelon. The authors used a competitive overcomplete output layer (COOL) to overcome the fooling in ML classification. Malware and benign classification using ML models may also suffer from such an anomaly where an outlier is put into the wrong category.

Harang and Ducau [6] used the concept of COOL that avoids fooling the ML classification to measure and visualize the concept drift of malware features with time. Concept drift is a change in input data to the ML model due to new malware samples over time and exclusion of old samples. The change in input results in retraining the ML model and consequently changing the parameters of the ML model. The authors focused their effort on 20,372 WannaCry samples and 138,429 HWorld samples of ransomware. They concluded that the individual malware families are stable with lower change from month to month for their feature representation. In addition, they suggested that the machine learning model evading is not used widely by malware authors.

Ceschin et al. [7] studied DDM and EDDM drift detection methods to detect changes in the malware over time. The drift detection methods help to determine whether a warning indication implies degraded machine learning models and confirmation by either DDM or EDDM method implies the need to immediately update the ML model. Jordaney et al. [8] proposed a Transcend framework for identifying aging in a ML classification models. The framework’s goal is to detect the performance degradation of the model much before deployment. They used concept drift, a statistical comparison of samples during deployment with those samples with which the model is trained. Their framework can raise flags when the model starts to make consistently poor decisions. Gove et al. [9] shared “SEEM”, a visualization solution, for the analysis of malware to compare the behavior of a large set of malware samples about the imported DLLs and API used as callback domain. Wagner et al. [10] used a knowledge-assisted visualization solution for malware analysis by Malheur tool to cluster system and API calls to find a pattern for malware. They used the help of domain experts to find the pattern and externalize the rules. These rules are used to find the malware in hope that similar patterns will be executed. This solution depends on old rules that may fall short of new rules embedded in future zero-day malware that are difficult to detect.

Pohl [1] described the vulnerabilities in the application, firewalls, operating system, anti-virus products, and hardware that can be discovered systematically with required tools or accidentally by users. These vulnerabilities must be tested with attack software or exploits. The life cycle of vulnerabilities can be described in three phases. In the first phase, the product does not have any vulnerability. In the second phase, a vulnerability may be found in software products. In the third phase, the vulnerability must be fixed and a patch for software has to be issued. It further distinguishes between zero-day vulnerability and less than zero-day vulnerability.

Ye et al. [11] used zero-day vulnerabilities to identify four security metrics for attack complexity and impact of a zero-day vulnerability. The four metrics are K-zero day safety, minimum zero-day vulnerability attack path length, the exploitability of zero-day attack path, and risk of zero-day attack path. K-zero day safety metric is the minimum number of zero-day vulnerabilities required to exploit the target and other metrics are associated with the difficulty of identifying the attack path length and the probability that a hacker may follow the path. The attacker may follow an unsecured path or a zero-day attack path. The fourth metric determines the risk of following a path.

Malware can be detected by one of the three methods such as static analysis, dynamic analysis, and hybrid analysis. In static analysis, the features of malware may be extracted from the PE header [12] or the Application Program Interface (API) calls from the loaded dynamic link library (DLL) [13]. Features for static analysis can also be extracted from software files, such as histogram of bytes in the sample, the entropy of parts of the sample file, and printable strings with more than five characters embedded in the sample file [14]. Raff et al. [15] used n-grams from byte code for static analysis. Further, n-grams can also be derived from assembly code, API calls, etc. The authors in [16] used the image of the file to perform machine learning and/or deep learning to identify the patterns in an executable to detect malware. The malware is not executed in static analysis. Static analysis has its shortcoming in malware detection if the malware is obfuscated by non-standard packing and encryption methods.

The execution of malware has to be in a virtual machine or by specific tools that use a protected environment. In the absence of a protected environment, the malware can infect the system and make the computing resource unusable. Tang et al. [17] observed that dynamic analysis can overcome the shortcomings of static malware analysis. Jindal et al. [18] executed the malware in a virtual environment and noted the actions performed by malware such as changes in the registry, files, processes, system configuration, and communication in the network. The dynamic analysis also has its shortcoming as malware authors try to uncover the virtual environment and put an end to the actions of the malware. This behavior leads the security analyst to believe the sample is benign software. In [19], the authors used hybrid analysis, which combines the technique of static analysis and dynamic analysis, to defeat their shortcoming. As the technique is combined, it has the same issues as dynamic analysis of being expensive from a resource perspective and static analysis from packing and encryption.

Higher prediction accuracies by our ML model give us motivation to further improve the malware detection by finding and proposing novel ways to visualize the top significant features of malware and the amount of contribution they have on a probability scale for classification and misclassification by the ML model. We go a step further to visualize why the prediction by an efficient ML model has minor deficiencies that lead to misclassification such as FP and FN and how the top nine features change along with the amount of contribution on a probability scale.

## 3. Methodology

### 3.1. Zero-Day Malware Model

We used the static malware analysis method for malware detection in this work. In static analysis, the features of the sample are taken. The features can be categorized into two major parts. Part one features consist of a Portable Executable (PE) header [20,21] and part two features consist of features derived from the file [12]. The PE header shown in Figure 1 has a DOS header, DOS Stub, and PE header. The PE header has many parts such as the General File header, Header file information, optional header, section headers, and many directories. The directories include but are not limited to the Import directory, Resource directory, Export directory, Exception directory, etc. The malware authors use custom tools to build an executable that results in inconsistent values in the PE header. These inconsistent fields can help distinguish malware and benign software. Hence, the fields from the PE header are included as features. These features can also be visualized using CFFexplorer. The import directory has a list of Dynamic Link Libraries (DLL) and API used by the software. Some executables export functions for sharing a function call with other programs. The resource directory has a list of icons, bitmap images, menus, strings, dialogs, configuration files, version information, etc. Malware uses specific functions in DLL to achieve its objectives. The imports and export function can help distinguish malware and benign software. Hence, these are included as features. The malware authors like to use specific icons, bitmap images, and strings to identify their groups. Those strings and icons can be found in the resource directories. The exception directory has a list for handling exceptions.

Part two of the features derived from the program are a histogram of bytes in the executable, strings in the file, and the entropy of the executable. The histogram of bytes in the sample is the count of byte values that range from 0–255. It helps to get the amount of availability of various bytes in the sample and good information for various known packing methods used by benign executable authors and both known and private packing methods used by malware authors to hide the malware code. These techniques increase the difficulties of malware analysts to get more insight into the malware. The packing and encryption methods raise the entropy of bytes in the executable. The second largest features are derived from the entropy of the file in a block size of 2048 bytes. This entropy is computed for the 2048 byte block size of a file with a step size of 1024. The values are in 16 × 16 bins representing the entropy and byte values as the method outlined in [15] by Saxe et al. The third largest features are derived from strings embedded in the executable file. Strings make very important information in the malware and lead to finding the command and control URLs, IP addresses for communication with malware authors, the signature of malware authors, and groups. In addition, these strings also represent the path of files, registries that are created, deleted, and modified. Strings of the size of five printable characters or more are hashed into 104 bins. A particular string after a hash will land in a particular bin only.

Hence, a specific URL, an IP address, a file name, and a signature will be mapped to a specific bin among 104 string features. These bins represent features. The part one and part two features are 2351 features for each sample.

We filtered, processed, and extracted three datasets, D1, D2, and D3, from a dataset shared by [12] as shown in Figure 2. The extracted datasets represent samples from different periods. Dataset D1 has malware and benign samples from January 2017. Dataset D2 has samples from February 2017. Dataset D3 has samples from March 2017.

We used a bagging and boosting machine learning algorithm model in the sklearn library as shown in the system block diagram in Figure 3. For bagging, Random forest and Extratree forest model and for boosting, gradient boost decision tree based LightGBM and XGboost models were used in this work. Machine learning models using each of the methods such as Random forest, Extratree, LightGBM, and XGBoost were made using dataset D1, which has samples from January 2017. These trained models were used to predict the future zero-day malware and benign samples from February 2017 and March 2017 represented by the D2 dataset and D3 dataset, respectively. All of the samples in D1, D2, and D3 have different SHA256 values and the D2 and D3 samples are not exposed to training ML models. A few top features in a range of 6–25 features among the 2351 features that help identify the malware and benign samples were visualized using various diagrams such as bar plot and waterfall plot. In addition, these diagrams show the amount of contribution by each top feature in shap value and scale of probability. Any machine learning model learns from a set of given samples, termed as training samples, and can be used for predicting the unseen samples. In this work, the unseen samples were malware and benign samples from the future period such as February 2017 and March 2017, represented by D2 and D3 datasets. Zero-day samples in this work were from the future period, had different SHA256 values, were not exposed, and were unknown to training of ML models. The prediction results are discussed in Section 4 and are very promising.

The machine learning model has a minor deficiency in detecting the malware and can misclassify malware as benign, a false negative, or benign software as malware, a false positive. For good performance of the ML model, both the false positive and false negative should be minimized. A false positive creates lots of inconvenience to the users and brings down the efficiency and productivity of an organization. A false negative is more dangerous as the user is unaware of the ill effects of malware, thinking it to be benign software. One such false negative can create a lot of damage to the organization. Hence, it is important to identify FP and FN predictions and change them to the correct category. Bar plots and waterfall plots give the top features based on Shapley value of features. The plots can indicate the Shapley value for each feature. Detection of change in a few top features can correlate with misclassified FP and FN predicted categories. Misclassification can also be confirmed by a change in the contribution by top feature. Both changes in the top feature and change in the contribution are used to make inductive rules that can be used to put the correct classification on misclassified samples

A new ML model was trained using dataset D2 and used to predict the future zero-day malware, benign software from March 2017, represented by dataset D3. These models were also used to predict the malware dataset D1 from January 2017. Figure 3 shows these experiments in the lower part of the block diagram with dataset D2.

### 3.2. Feature Importance Visualization in Shap Value

An ML model should be both accurate and interpretable. Decision tree-based machine learning models can be interpreted based on decision path, information gain, and heuristic value to features. This work uses Shapley value as a heuristic value for each feature in the ML model to make it interpretable. A local explanation for a feature means to assign a numerical value or credit to each feature used in the ML model based on the tree. The credit for features used in the decision tree-based ML models in this work is Shapley value. This local explanation to a feature should also support global explanation for the tree. A global explanation is an overall prediction for a sample by the ML model. Shapley values are also used as the important feature for the tree-based ML model in this work. This requires that the feature importance should use a measure that has local accuracy, consistency, and missingness [22]. Shapely values from game theory satisfy all three properties simultaneously and can be used as a numerical value to a feature in the tree-based ML model.

Optimal local explanation based on Shapely value was implemented as TreeExplainer by Lundberg et al. [22]. A global structure can be made from the addition of local explanation. Hence, the overall prediction of the ML model can retain local faithfulness. The exact calculation of local explanation is NP-hard. However, the TreeExplainer by Lundberg et al. applies approximation to do the computation in polynomial time. TreeExplainer provides feature interaction of the tree-based ML model and can be captured by extended property of local explanation. Hence, the TreeExplainer provides valuable insight into the behavior of the ML model.

The Shapley values can help find the top features but cannot be correlated to predicted value by a ML model. Hence, this project converts the Shapley value to probability scale using Algorithm 1.

**Algorithm 1** Shap value to probability value

**Input**

$:\;\mathrm{Shap}\;\mathrm{Explanation}\;\mathrm{structure},\;{\mathrm{MLModel}}_{\mathrm{predict}}$



**Output**

$:\;\mathrm{Probability}\;\mathrm{value}\;\mathrm{for}\;\mathrm{each}\;\mathrm{feature}\;({\mathrm{probability}}_{\mathrm{feature}})$

1Shap value **= SBV**= Shap_explanation.basevalue;2Logit_base_value

$={\mathrm{logit}}_{\mathrm{bv}}=\frac{1}{{1+e}^{-\mathrm{SBV}}}$

**;**
3Distance between logit base value and model predicted value **=**
${\mathrm{Diff}}_{\mathrm{ML}}$$={\mathrm{MLModel}}_{\mathrm{predict}}{-\mathrm{logit}}_{\mathrm{bv}}$;4Coefficient **=**
${\mathrm{coff}}_{\mathrm{ML}}$$=\frac{\sum \mathrm{Shapvalue}}{{\mathrm{Diff}}_{\mathrm{ML}}}$;5**For** each shap value in shap explanation structure;6
Transform value= $T_{\mathrm{value}}$ = $\frac{\mathrm{Shapvalue}}{{\mathrm{coff}}_{\mathrm{ML}}}$;7
**end**
8

${\mathrm{probability}}_{\mathrm{feature}}$

$=T_{\mathrm{value}}$;

### 3.3. Conversion from Shap Value to Probability

Algorithm 1 gives the algorithm to convert the Shapley value for a sample to the probability scale. A shap explanation object is computed for each sample based on the shap library in python and the trained ML model. The shap explanation object is a data structure that has a base value and array of contributions of each feature in SHAP value. In step1, the base value of the shap_explanation object is extracted. The base value of the explanation object is converted to the logit base value based on the logit function identified in step2 of the algorithm. The logit base value is the base start point in waterfall plots. Subsequently, we compute the difference between the predicted probability values by machine learned model with logit base value. The computed difference is proportionately distributed over the feature as per the shap value of each feature. The proportionate value is the contribution of a feature on a scale of probability. The addition of this probability value will be equal to the predicted probability value of the ML model for the sample. The Shapley value of each feature is transformed to the probability scale using the algorithm in Algorithm 1. The probability scale helps identify top features.

For good performance of the ML model, both the false positive and false negative should be minimized. The false positive creates lots of inconvenience to the users and brings down the efficiency and productivity of an organization. The false negative is far more dangerous as a user is unaware of the ill effects of malware. They think of the software as benign and use it. One such false negative can create all the damage to the organization. FP, FN, TN, and TP samples from the best performing ML model were used to further improve the performance of the ML model. We studied the trends in the top features for all predicted categories. This trend helps to form inductive rules. The inductive rules may be applied to a zero-day malware for effective and robust prediction. The prediction results are discussed in Section 4.

## 4. Experimental Results and Analysis

### 4.1. Datasets

Dataset D1, D2, and D3 were extracted from [12] by filtering and processing as per Figure 2. Samples were selected for a dataset based on a specific period. The samples in the D1 dataset are from January 2017. The counts of unlabeled, malware, and benign samples are described in Table 3. The samples in the D2 dataset are from February 2017 and the samples in D3 are from March 2017. All the samples in D2 and D3 have different SHA256 values compared to samples in the D1 dataset, future period, and are not exposed to training the ML models. The counts of samples in unlabeled, malware and benign are in Table 3. The unlabeled samples were excluded from experiments in this work. Without labels, the unlabeled samples cannot be ingested for training or testing considering the supervised ML algorithms planned for this work. Each sample in the dataset has 2351 features as described in Section 3.1. Table 4 lists the names of 2351 features, counts of features, and parts of the sample file from where the features are derived. It has 256 features from the histogram property of the file and the names of features are H1-H256. It has 256 features, named BEn1-BEn256, from the entropy property of samples. These features show the entropy of the file. High entropy indicates the use of packing and encryption methods by the malware authors. Str1-Str104 are 104 hashed strings features from a sample. It contains all the strings that have five or more printable characters.

It has 10 features from the general file information part, 62 features from the header file information part, and 255 features from the section information part of the PE header of the sample file. These parts are shown in the PE header in Figure 1. The imports of API from DLL are 1280 features with the names Imp1 to Imp1280. It has 128 features, exp1 to exp128, from exports of API from samples. Figure 1 shows part of the sample file from where they were extracted.

### 4.2. Design of Experiments

A Core(TM) i5-2701 MHz laptop with 8 GB Ram was used for the experiment setup.

ML algorithms such as LightGBM and XGBoost were selected for zero-day malware detection. These ML algorithms were selected for the following advantages.

Insight into the feature importance of the model.Continuity of features with future upcoming malware.Change of features with future upcoming malwareEase of computationBetter performance of models

### 4.3. Zero-Day Malware Detection with XGBoost

The XGBoost ML model was trained with the 70% samples from the D1 dataset and tested with 30% samples from the D1 dataset. Table 5, Row1 has the results of the test. The “D2-D1” row in the table gives performance data for the D2 dataset representing zero-day malware from February 2017 using the XGboost model trained with the D1 dataset. The “D3-D1” row in the table gives performance data for the D3 dataset representing zero-day malware from March 2017 using the XGboost model trained with the D1 dataset. We can see minor degradation in accuracy of future malware from February 2017 (98.50% vs. 97.87%). We see further degradation in accuracy to 97.50% with future zero-day malware from March-2017. This experiment was repeated with training the XGBoost model with the D2 dataset from February 2017 and predicting the zero-day malware from the D3 dataset from March-2017. The performance results are in row “D3-D2”. We also predicted the old malware from January 2017 in the D1 dataset with this model and performance results are in row “D1-D2” row.

The number of false positives (FPs) increased from 142 in D1-test to 914 with the D2 dataset (an increase of 2.56% for the test dataset from D1 to 2.86% for the D2 dataset). The FPs further increased to 806, representing an increase to 3.29%. The False negatives increased from 105 to 362 for the D2 dataset, representing an increase from 0.97% for the test dataset from D1 to 1.32% for the D2 dataset and a minor decrease to 140 with the D3 dataset, representing a decrease to 1.10%.

### 4.4. Zero-Day Malware Detection with LightGBM

The experiment as outlined in Section 4.3 for the XGBoost GBDT model was repeated with the LightGBM ML algorithm. The results of the experiment are in Table 6. It has results for Accuracy, confusion matrix parameters, Precision, Recall, and F1-score. The accuracy performance of the D1-test and D2 dataset was compared and we noticed minor degradation from 98.48% to 97.64%. The performance between the D1-test and D3-D1 row shows further minor degradation in accuracy to 97.20% with the D3 dataset.

The F1-score degraded with the D2 dataset from February-2017 from 0.99 to 0.97 and it further degraded to 0.96 with the D3 dataset containing future malware from March-2017. The false positives increased from 150 to 989 with the D2 dataset (an increase of 2.71% to 3.10%). This further increased to 920, representing an increase to 3.77%. The False negatives increased from 100 to 428 for the D2 dataset, representing an increase from 0.92% to 1.57% and a minor increase to 142 with the D3 dataset representing an increase to 1.12%.

### 4.5. Zero-Day Malware Detection with Random Forest

The experiment as outlined in Section 4.3 and Section 4.4 for boosting GBDT ML algorithm was repeated for bagging ML algorithm Random Forest. The results are in Table 7. Minor degradation in accuracy of future malware from D2 dataset (96.72% vs. 97.76%) was found. We saw further degradation in accuracy to 96.13% with future malware of the D3 dataset. This is slightly less than the XGboost model.

The False positives (FPs) increased from 186 to 1283 with the D2 dataset (an increase of 3.39% for the test dataset from D1 to 4.06% for the D2 dataset). This was computed by the ratio of FP to TP. The FPs further increased to 1228, representing an increase to 5.10%. The False negatives increased from 183 to 682 for the D2 dataset, representing an increase from 0.97% for the test dataset from D1 to 1.69% for the D2 dataset and an increase to 239 with the D3 dataset representing an increase to 1.88%. This was computed by FN to support, which represents the total malware present in the dataset.

### 4.6. Zero-Day Malware Detection with Extratree

The Extratree ML model was trained with the 70% samples from the D1 dataset and tested with 30% samples from the D1 dataset. Table 8, Row1 has the results of the test. We used the model trained on the D1 dataset to predict the malware and benign samples from datasets D2 and D3. D2-D1 row in the table gives performance data for dataset D2 using the Extratree model made with the D1 dataset. The “D3-D1” row in the table gives performance data for the D3 dataset representing March 2017 using the Extratree model with the D1 dataset. We could see minor degradation in accuracy of future malware from February 2017 (97.96% vs. 97.35%). We did not see any degradation in accuracy with future malware of March 2017.

The False positives (FPs) increased from 166 to 1022 with the D2 dataset (an increase of 3.01% for the test dataset from D1 to 3.21% for the D2 dataset). The FP further increased to 1059, representing an increase to 4.37%. The False negatives increased from 169 to 568 for the D2 dataset, representing an increase from 1.56% for the test dataset from D1 to 2.08% for the D2 dataset and a minor decrease to 181 with the D3 dataset, representing a decrease to 1.42%.

### 4.7. Comparison of Model Performance

Table 9 compares the accuracy among the four machine learning models LightGBM, XGBoost, Random Forest, and Extratree for being trained on D1 and zero-day malware prediction of D2 and D3 datasets. We found the highest accuracy with XGBoost and the lowest with Random forest for a model trained on dataset D1 and zero-day malware prediction on D2 and D3 datasets. For models trained on the D2 dataset and zero-day malware, prediction was done for the D3 dataset. The XGBoost model gave the consistently highest result and the Random Forest model gave the lowest accuracy.

Table 10 compares the false positives and false negatives for the LightGBM (identified as LG), XGBoost (identified as XG), Random Forest (Identified as RF), and Extratree (identified as ET). The lowest false positives were detected by the XGBoost model and the highest by Random Forest. A model should have minimal false positive and false positive detection. Hence, the XGBoost model was selected for further improvement to the model.

### 4.8. Improvement to XGBoost Model by Trend in Top Features

The complete dataset D2 is a combination of malware and benign samples as detailed in Table 3. The D2 dataset was used for future zero-day malware prediction using the XGBoost model trained from the D1 dataset. Based on the prediction of the ML model, samples that are misclassified in FP and FN categories are identified. Samples that are classified correctly in TN and TP are also identified. Figure 4 displays the top 25 features with names in bar plots in each category of FP, FN, TN, and TP. Only the top twenty five features were selected as the diagram is clear and their contribution was easier to observe. Figure 4 displays the contribution of the top 25 features in shap value in bar plots in each category of FP, FN, TN, and TP. Only the top twenty five features were selected as the diagram is clear and their contribution was easier to observe. The figure shows bar plots of a FP sample (first row left), a FN sample (first row right), a TP sample (second row left), and a TN sample (second row right) in shap value. Figure 5 shows bar plots for FP, FN, TN, and TP samples on a probability scale.

The XGBoost ML model predicts the probability for a new sample. If the probability is more than 0.5, then the sample under test is termed malware, otherwise it is termed benign. It also displays the contribution of the top 25 features on a probability scale. The probability values are determined from the shap value as per the algorithm in Algorithm 1. The feature names can be referred to in Table 2. The contributions of the remaining 2326 features are displayed in the last bar of the figure.

Figure 6 shows waterfall plots for a sample in each category of FP, FN, TN, and TP as predicted by the trained XGBoost ML model. The waterfall plot identifies the top nine features with name and their contribution in shap value. The feature name can be referred to in Table 2. Figure 7 shows waterfall plots for a sample in each category of FP, FN, TN, and TP on a probability scale. The probability scale value is computed using the algorithm in Algorithm 1 from shap values of top features. The last bar in the waterfall plot shows the contribution of the remaining 2346 features out of 2351 features. All of the feature’s probability values can be summed to the predicted value by the ML model. The top left waterfall plot in Figure 7 is for a FP sample. The sum of all features is 0.659 (more than 0.5) and the label for the sample is benign. Hence, the prediction is FP. The top right waterfall plot in Figure 7 is for a FN sample. The sum value is 0.205 (less than 0.5) and the label for the sample is malware. Hence, the prediction is FN. Figure 7 lower row waterfall is for a TN and TP prediction by the ML model with sum value 0.013 and 1.

The waterfall plots for a false positive sample in Figure 6 and Figure 7 display the top features that contribute in both negative and positive directions compared to the contribution of 2342 features. The observation is the same for the top features in the waterfall plot for a FN predicted sample compared to the contribution of 2342 features for a false negative sample. However, the waterfall plot for a TN predicted sample shows the top features that contribute in negative directions as the contribution of 2342 features in a negative direction. The observation is the same for the true positive sample. The top features contribute in a positive direction and are in line with the contributions of 2342 features. These observations lead us to conclude that the waterfall plots in Figure 6 and Figure 7 for false negative and false positive categories are very different from waterfall plots for TP and TN categories. The following two important conclusions are noted:The top features contribute to different directions than 2342 features for FP and FN samples.The top features contribute to the same direction as 2342 features for TP and TN samples.

This observation and conclusion can be used for the correct classification of misclassified samples. The FP and FN prediction can be identified using waterfall plots. The waterfall plot for the FP and FN categories will show that the top features for the sample will contribute in both positive and negative directions compared to 2342 features. The TP and TN samples can be identified using their waterfall plots. The waterfall plots for TP and TN categories will display the top features that will contribute in the same direction as 2342 features.

### 4.9. Derivation of Inductive Rules

The five top features are identified in Table 11 in the features column for FP, FN, TN, and TP samples. It identifies the five top features with “P” and the amount of contribution of a feature on the probability scale for each category of prediction. The contribution of the topfeature for each category is in bold, such as C_Char1 (0.03) for the TP category and Imp321 (−0.18) for the FP category. A top feature present in a category of the sample such as Imp321 in FP but not present in other categories such as FN, TP, and TN is marked as “N”. Few cells have feature names identified such as in row 13. The “Features” column has the value BEn253, but the “false positive” column has the value BEn50. This means that the FP sample does not have any contribution from the BEn253 feature but has a contribution from the BEn50 feature. The following observations can be derived from the table:FP, FN, TP, and TN predicted samples have a disjoint set of features. The top features are very different for each predicted sample category.The top feature for a false negative sample is “Rx_sec_num” and is present with a very low contribution in the false positive and true negative.The top feature for true positive is “C_char1” and contributes very low, but has a high contribution for samples in FP and FN.The probability scale value is high for 2342 features for the TP sample compared to other top features.

Having identified a misclassified prediction by a waterfall plot by trend, the technique outlined with data in Table 11 can be used to differentiate between FP and FN prediction. Waterfall plots can also be used for confirmation of the correct classification of samples.

The Inductive rules are derived as follows.

An unknown real-time sample that has top features of Imp321, C_char1, H33, and Str43 is a FP sample. The amount of contribution from 2342 features is opposite to Imp321. It is predicted as malware but can be used as benign software.An unknown real-time sample that has the top features of Rx_sec_num and the contribution of 2342 features is opposite to Rx_sec_num features is a FN sample. It is predicted as benign but is actually malware. Manual static and dynamic analysis of the sample may be performed.An unknown real-time sample that has C_char1 as the top feature, no contribution of Imp321 feature, and much higher contribution from the remaining 2342 features is malware. It is a verification of the sample and a robust prediction.An unknown real-time sample that has the top feature of Subsystem9 and contribution from 2342 features in the same direction as Subsystem9 should be a benign sample. It is a verification of the sample and a robust prediction.

As all the misclassified samples can be corrected by the trends in top features, the result of the ML model for XGBoost reaches 100% accuracy for the two datasets D2 and D3 in a controlled environment. Please note this is not a property of ML algorithms but enhancement due to the trend in top features identified using Shapley values. Hence, all the objectives (bullet points listed in the Section 1) are met.

### 4.10. Comparison with Five Zero-Day Malware Projects

Table 12 has comparison details of this work with five zero-day malware detection projects. This project achieves higher accuracy with a large number of future zero-day malware from D2 and D3 datasets compared to Yousefi-Azar et al. [4]. The Source of unknown malware is not specified in Venkatraman et al. [3]. It may be possible they also took future malware as unknown malware. Jung et al. [23] took 333 known malicious adobe flash files with the extension .swf and 333 benign .swf for training the ML model, and tested it with .swf files from 2007–15 as zero-day malware. They achieved 51–100% accuracy. Alazab et al. [24] considered all unknown malware as zero-day malware and achieved an accuracy of 98.6. This work has an accuracy of 98.49%. They used much higher malware (three times) compared to benign software for training. They created new malware using code obfuscation that changed the code of malware but did not change the functionality of the malware. This is the same as malware generation using polymorphic and metamorphic engines. This work used 256 entropy features of the sample file and 50 features from the entropy of sections of the file. These 256 entropy features were extracted by dividing the sample file into multiple parts of size 2048 bytes with an overlapping window for 1024 bytes and then taking the entropy of each part. These 306 features can detect such obfuscation in sample malware files. Shafiq et al. [25] used the 10K malware dataset from Vxheavens, 5 K malware from malfease dataset (now not accessible), and 1447 benign files from a local laboratory and divided them into malware types such as backdoor, DoS, Nuker, Trojan, Virus, etc. They stated that unknown malware or malware with unknown signatures are zero-day malware. However, they did not describe ways to determine unknown malware or malware with unknown signature but the test samples were drawn from Vxheaven and malfease datasets.

There is much research work related to a zero-day attack. A zero-day attack refers to a new network signature for Denial of Service (DOS) or Distributed Denial of Service (DDOS). Kumar et al. [26] stated that zero-day attacks are a comparison between genuine network data to find heavy hitters in attack data. A heavy hitter is to find a new signature responsible for DOS or DDOS in network traffic data. They claimed that they found the heavy hitters in a low amount of network data. Kim et al. [27] described that zero-day DDOS attacks on websites are preceded by fingerprinting and infection of hosts or devices. They captured network traffic to detect such early attacks and to detect emerging botnets and new vulnerabilities. They monitored at the port level on a different section of the network to detect anomalies and thereby new DDOS attacks. Such works are not considered for comparison as the objective to find zero-day malware does not match.

## 5. Conclusions

In this work, three datasets, D1, D2, and D3, from Jan, Feb, and March 2017, respectively, were derived from [12] and used for zero-day malware prediction. Static analysis was used to get 2351 static features from the PE header and properties of files. Bagging and boosting ML models were trained using samples from the D1 dataset, which had 2351 features. The trained ML model was used to predict samples in datasets D2 and D3 to detect zero-day malware. The best model among the bagging and boosting ML model was selected based on accuracy and lowest FP and FN predictions. The best performance was consistent from XGBoost at 97.87 and 97.50 accuracy, respectively, from D2 and D3 dataset as future zero-day malware. The XGBoost model using dataset D2 from February 2017 was used to predict the future malware in dataset D3 and the performance gave 98.492 accuracy. Bar plots and Waterfall plots based on a contribution of features in shap value and probability scale were displayed. The plots helped identify the top features. Top features were compared for FP, FN, TN, and TP categories of samples. The top features and their contribution are different for each predicted category. The top features and their contribution can be used to identify the FN and FP misclassified category of prediction. The comparison among each category of the sample demonstrated trends in the top features. These trends were used to detect misclassification in FP and FN samples. Hence, all the misclassified samples can be put in correct categories leading to an increase in efficiency of the ML model. The trend can also be used for confirmation of TP and TN samples for robust detection of unknown zero-day malware and differentiate them from misclassification.

Future work in this area may be to train ML models for a specific family of malware such as Trojan horse, Rootkit, Ransomware, etc. using a real-time large dataset. This could help find reasons for misclassification and reduce the misclassification counts.

## Figures and Tables

**Figure 1 sensors-22-02798-f001:**
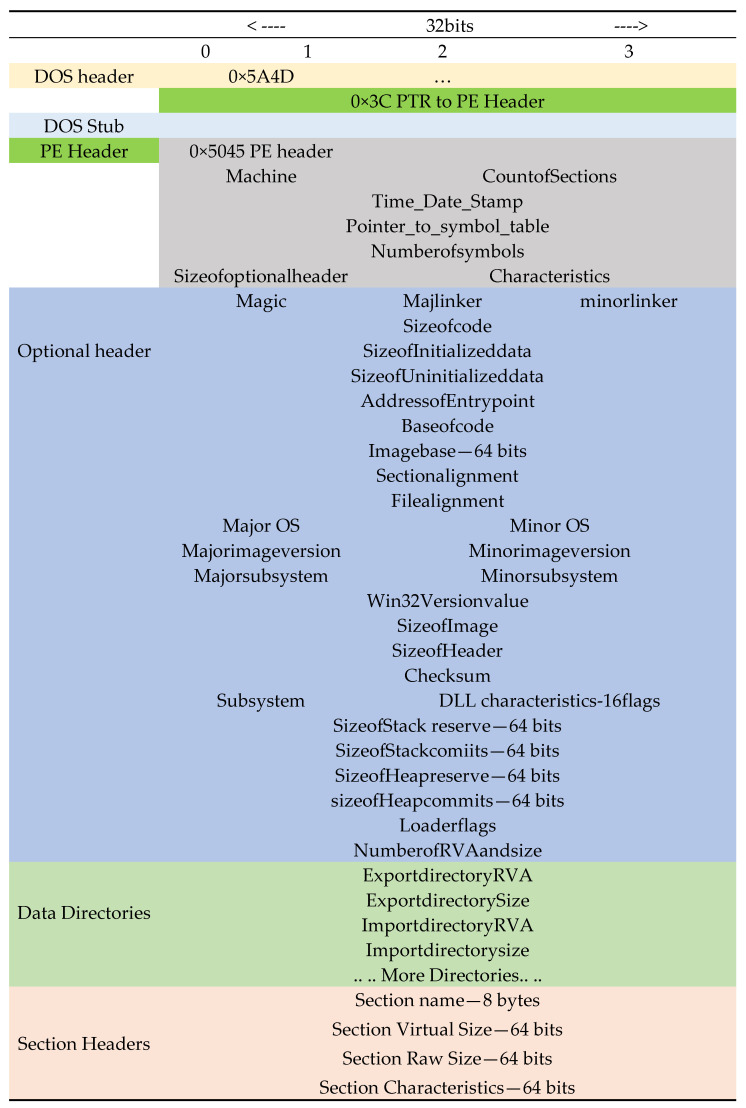
File header of a sample with details of components in the PE header.

**Figure 2 sensors-22-02798-f002:**
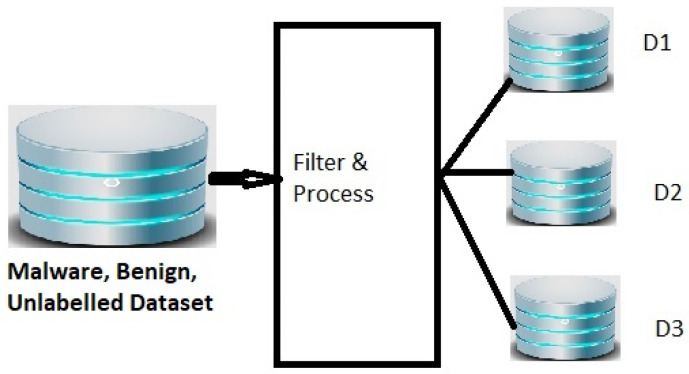
Derivation of D1, D2, D3 datasets from the Malware dataset scale.

**Figure 3 sensors-22-02798-f003:**
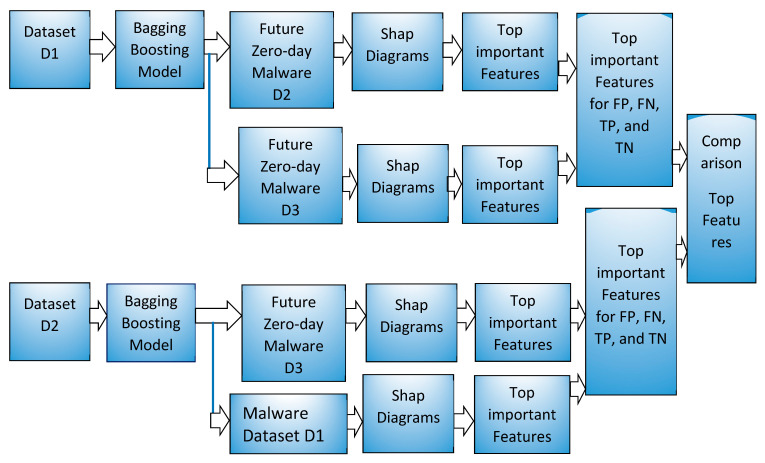
Block diagram for the method.

**Figure 4 sensors-22-02798-f004:**
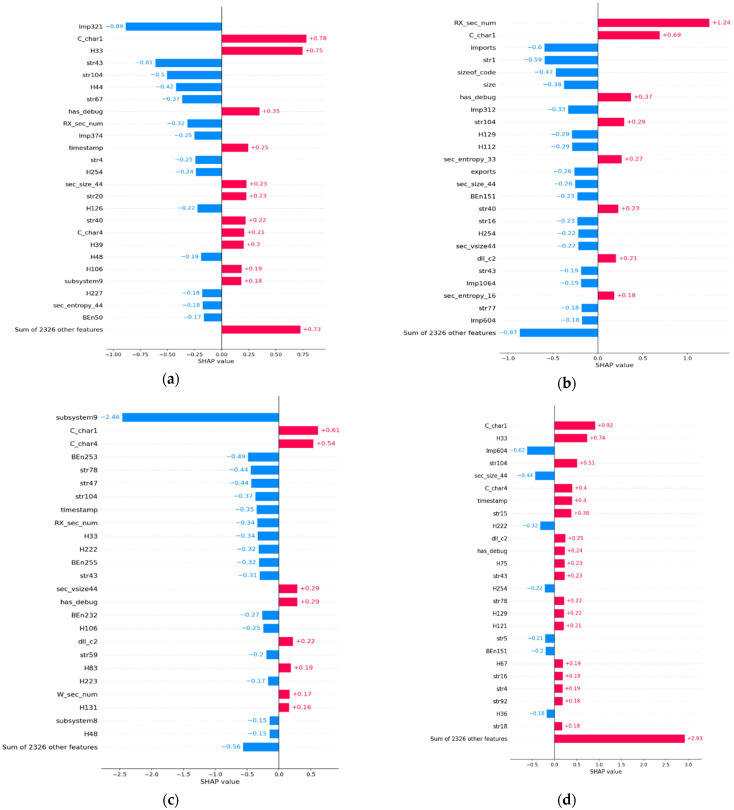
Bar plot of a FP sample (**a**), a FN sample (**b**), a TP sample (**c**), and a TN sample (**d**) from the D2 dataset in shap value.

**Figure 5 sensors-22-02798-f005:**
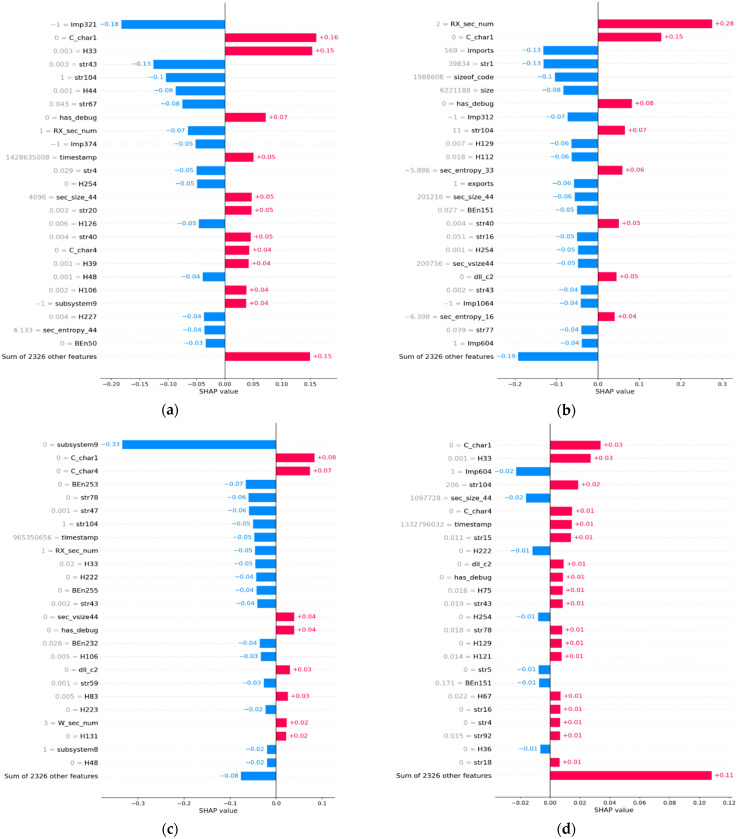
Bar plot of a FP sample (**a**), a FN sample (**b**), a TP sample (**c**), and a TN sample (**d**) from the D2 dataset on probability scale.

**Figure 6 sensors-22-02798-f006:**
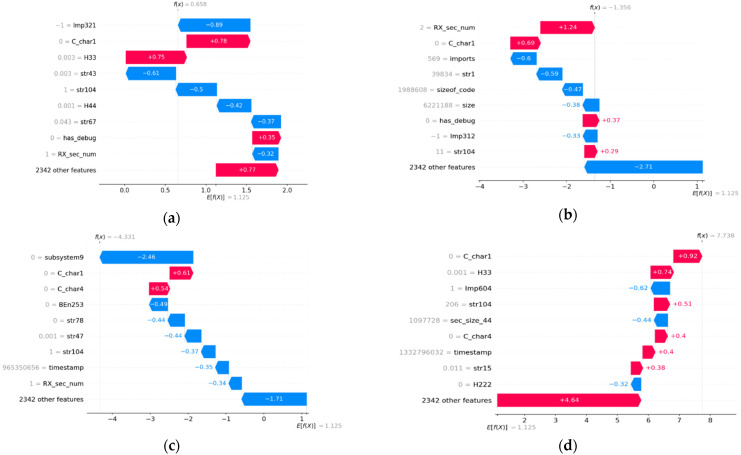
Waterfall plot of a FP sample (**a**), a FN sample (**b**), a TP sample (**c**), and a TN sample (**d**) from the D2 dataset in shap value.

**Figure 7 sensors-22-02798-f007:**
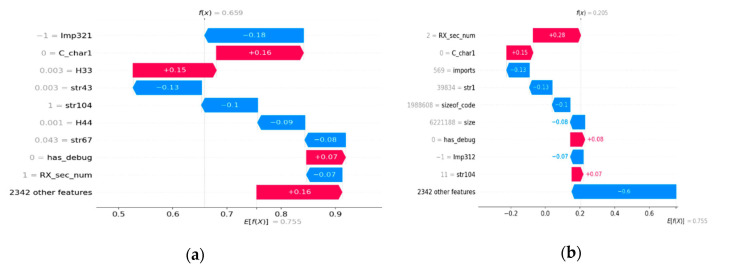
Waterfall plot of a FP sample (**a**), a FN sample (**b**), a TP sample (**c**), and a TN sample (**d**) from the D2 dataset on probability scale.

**Table 1 sensors-22-02798-t001:** Top ten software producers with vulnerabilities.

Sl. No.	Vendor Name	Count of Products	Count of Vulnerabilities	#Vulnerabilities/ #Products
1	Cisco	5623	4159	1
2	IBM	1335	5378	4
3	Oracle	971	8270	9
4	Microsoft	665	8391	13
5	Redhat	430	4058	9
6	Apple	140	5467	39
7	Google	128	6916	54
8	Debian	109	6022	55
9	Canonical	49	3180	65
10	Fedora Project	21	2885	137

**Table 2 sensors-22-02798-t002:** Top Operating systems with Vulnerabilities.

Sl. No.	Product Name	Vendor Name	Count of Vulnerabilities
1	Debian Linux	Debian	5572
2	Android	Google	3875
3	Ubuntu Linux	Canonical	3036
4	Mac Os X	Apple	2911
5	Linux Kernel	Linux	2722
6	Fedora	Fedoraproject	2538
7	iphone OS	Apple	2522
8	Windows 10	Microsoft	2459
9	Windows Server 2016	Microsoft	2233
10	Windows 7	Microsoft	1954

**Table 3 sensors-22-02798-t003:** Details of D1, D2, D3 datasets.

Sl. No.	Short Name	Counts	label	Period
1	D1	28,606	Unidentified	January 2017
2		17,180	Benign	January 2017
3		32,761	Malware	January 2017
4	D2	31,394	Unidentified	February 2017
5		32,820	Benign	February 2017
6		27,239	Malware	February 2017
7	D3	15,656	Unidentified	March 2017
8		25,261	Benign	March 2017
9		12,692	Malware	March 2017

**Table 4 sensors-22-02798-t004:** Features names in the dataset used.

Description	Count	Feature Name		
Histogram	256	H1–H256		
Byteentropy	256	BEn1–BEn256		
String Extractor	104	Str1–str104		
General file Info	10	size		
		vsize		
		has_debug		
		exports		
		imports		
		has_relocations		
		has_resources		
		has_signature		
		has_tls		
		symbols		
Header file Info	62	**Feature names**	**Num of features**	**Description**
		timestamp	1	Timestamp
		Machine1–Machine10	10	H/W type hashed
		C_char1–C_char10	10	Characteristics Hashed
		subsystem1–subsystem10	10	Subsystems Hashed
		dll_c1–dll_c10	10	DLL Characteristics hashed
		magic1–magic10	10	Magic
		major_i_verminor_i_ver	1	Image version
			1	
		major_linker_verminor_linker_ver	1	Linker version
			1	
		major_os_ver	1	OS versions
		minor_os_ver	1	
		major_ss_ver	1	Subsystem version
		minor_ss_ver	1	
		sizeof_code	1	Code size
Section info	255	Feature names	Num of features	Description
		num_of_sec,	1	Number of Sections
		num_of_sec_morethan0,	1	Number of sections >0
		num_sec_no name,	1	Count of sections without name
		RX_sec_num,	1	Count of sections with read and execute permission
		W_sec_num	1	Number of sections with write permission
		sect_size_1–sect_size_50	50	Section size hashed
		sect_entropy_1–sect_entropy_50	50	Section Entropy hashed
		sect_vsize1–sect_vsize50	50	Section vsize hashed
		entry_name1–entry_name50	50	Section name hashed
		sect_char1–sect_char50	50	Section characteristics hashed
Imports	1280	Imp1–Imp1280		
Exports Info	128	exp1-exp128		

**Table 5 sensors-22-02798-t005:** Results of XGBoost ML model.

	Accuracy	TP	FP	FN	TN	Precision	Recall	F1-Score	Support
**D1-Test**	98.501	5528	142	105	10,706	0.99	0.99	0.99	10,811
**D2-D1**	97.875	31,906	914	362	26,877	0.97	0.99	0.98	27,239
**D3-D1**	97.507	24,455	806	140	12,552	0.94	0.99	0.96	12,692
**D2-Test**	98.703	10,686	100	157	8877	0.99	0.98	0.99	9034
**D1-D2**	98.081	16,926	254	704	32,057	0.99	0.98	0.99	32,761
**D3-D2**	98.492	24,936	325	247	12,445	0.97	0.98	0.98	12,692

**Table 6 sensors-22-02798-t006:** Results of the LightGBM ML model.

Dataset	Accuracy	TP	FP	FN	TN	Precision	Recall	F1-Score	Support
**D1-Test**	98.483	5520	150	100	10,711	0.99	0.99	0.99	10,811
**D2-D1**	97.640	31,831	989	428	26,811	0.96	0.98	0.97	27,239
**D3-D1**	97.201	24,341	920	142	12,550	0.93	0.99	0.96	12,692
**D2-Test**	98.526	10,658	128	164	8870	0.99	0.98	0.98	9034
**D1-D2**	97.865	16,876	304	762	31,999	0.99	0.98	0.98	32,761
**D3-D2**	98.224	24,847	414	260	12,432	0.97	0.98	0.97	12,692

**Table 7 sensors-22-02798-t007:** Results of Random Forest ML model.

Dataset	Accuracy	TP	FP	FN	TN	Precision	Recall	F1-Score	Support
**D1-Test**	97.761	5484	186	183	10,628	0.98	0.98	0.98	10,811
**D2-D1**	96.728	31,537	1283	682	26,557	0.95	0.97	0.96	27,239
**D3-D1**	96.134	24,033	1228	239	12,453	0.91	0.98	0.94	12,692
**D2-Test**	97.815	10,642	144	289	8745	0.98	0.97	0.98	9034
**D1-D2**	96.776	16,869	311	1299	31,462	0.99	0.96	0.98	32,761
**D3-D2**	97.660	24,778	483	405	12,287	0.96	0.97	0.97	12,692

**Table 8 sensors-22-02798-t008:** Results of Extratree ML model.

Dataset	Accuracy	TP	FP	FN	TN	Precision	Recall	F1-Score	Support
**D1-Test**	97.967	5504	166	169	10,642	0.98	0.98	0.98	10,811
**D2-D1**	97.352	31,798	1022	568	26,671	0.96	0.98	0.97	27,239
**D3-D1**	96.732	24,202	1059	181	12,511	0.92	0.99	0.95	12,692
**D2-Test**	98.219	10,680	106	247	8787	0.99	0.97	0.98	9034
**D1-D2**	97.138	16,933	247	1182	31,579	0.99	0.96	0.98	32,761
**D3-D2**	98.232	24,920	341	330	12,362	0.97	0.97	0.97	12,692

**Table 9 sensors-22-02798-t009:** Comparison of Accuracy for LightGBM, XGBoost, Random Forest, and Extratree.

Dataset	LightGBM	XGBoost	RF	Extratree
**D1-test**	98.483	98.501	97.761	97.967
**D2-D1**	97.640	97.875	96.728	97.352
**D3-D1**	97.201	97.507	96.134	96.732
**D2-Test**	98.526	98.703	97.815	98.219
**D1-D2**	97.865	98.081	96.776	97.138
**D3-D2**	98.224	98.492	97.660	98.232

**Table 10 sensors-22-02798-t010:** Comparison of False Positives and False Negatives for LightGBM, XGBoost, Random Forest and Extratree.

Dataset	FP				FN			
	LG	XG	RF	ET	LG	XG	RF	ET
**D1-test**	150	142	186	166	100	105	183	169
**D2-D1**	989	914	1283	1022	428	362	682	568
**D3-D1**	920	806	1228	1059	142	140	239	181
**D2-Test**	128	100	144	106	164	157	289	247
**D1-D2**	304	254	311	247	762	704	1299	1182
**D3-D2**	414	325	483	341	260	247	405	330

**Table 11 sensors-22-02798-t011:** Comparison of features and their probability scale value in all predicted categories by XGBoost model. Bold means topmost feature.

Sl. No.	Features	Features Included in XGBoost ML Prediction Categories and Their Contribution			
		False Positive (FP)	False Negative (FN)	True Positive (TP)	True Negative (TN)
1	Imp321	**−0.18**	N	N	N
2	C_char1	P, 0.16	P, 0.15	**0.03**	P, 0.08
3	C_char4	P, 0.04	N	P, 0.01	P, 0.07
4	H33	P,0.15	N	P, 0.03	N
5	Subsystem9	0.04	N	N	P, **−0.33**
6	Rx_sec_num	−0.07	P, **0.28**	N	+0.05
7	Str43	P,−0.13	N	P, 0.01	P, −0.04
8	Str104	P, −0.1	P, 0.07	0.02	−0.05
9	Imports	N	P, −0.13	N	N
10	Str1	N	P, −0.13	N	Str78. Str47
11	Sizeof_code	N	P, −0.10	N	N
12	Imp604	N	N	P, −0.02	N
13	BEn253	BEn50	N	BEn151	−0.07
14	Other 2342 features	P, 0.16	P, −0.60	P, **0.17**	**−0.23**

**Table 12 sensors-22-02798-t012:** Comparison with five zero-day malware projects.

Project	Methodology	Details of Dataset	Result/Accuracy	Consideration for Zero-Day Malware
This work	Boosting algorithms: LightGBM, XGBoost Bagging algorithms: Random Forest, Extratree	Three datasets D1, D2, D3 of January 2017, February 2017, March 2017 With D1 for training and D2, D3 for prediction D2 for training and D3 for prediction and effect on the D1 dataset Details in Table 3	98.49	Train with the dataset from January 2017 and predict for malware from February 2017, March 2017. Train with February 2017 and predict malware from March 2017
Yousefi-Azar et al. (2018)	natural language processing and the term frequency tf-simhasing: multiply random projection matrix with term frequency of sample	Android: Drebin (5555 Malware, 5555 Benign software) DexShare (20,255 Malware, 20,255 Benign software) Windows PE files: 2016: Training dataset 11,983 Malware, 8912 Benign software 2017: Testing dataset 12,127 Malware, 11,983 Benign software	97.33	Dataset from 2016 to build the model and dataset from 2017 as zero-day malware to test against the model.
Venkatraman and Alazab (2018)	1a. Kernel Function 1b. Make images from malware files and use them as input to CNN (pre-trained) to derive features 1c. Plot features into t-SNE to visualize clusters. 1d. Use of K-means for further clustering using image features. 2. SVM with SMO-Normalized Polynomial	52 k samples	98.6%	Unknown Malware
(Jung & Kim, 2015)	1. Extract API call sequence features from both static analysis and dynamic analysis. 2a. Use static analysis feature with Deep Feed-forward Neural Network, Recurrent Neural Network for 2b. Use dynamic analysis feature with Recurrent Neural Network	Benign 333 (.swf files) Malicious 333 (.swf files)	51% to 100%	2007 to 2014 dataset for training and 2015 dataset for testing
(Alazab et al., 2010)	k–Nearest, Neighbor (kNN), Naïve Bayes (NB), Sequential Minimal Optimization (SMO) algorithm with (SMONormalized PolyKernel, SMOPolyKernel, SMOPuk, and SMO-RBF) kernels.	Benign software 15,480 Malware 51,223	98.6	1. Unknown malware 2. Unknown signature. New malware created using code obfuscation techniques with same functionality.
(Shafiq et al., 2015)	Ripper and SVM-SMO classifier	Benign software 1447 Malware 14,478	99.2% g AUC	Malware whose signature is not in database/Unknown malware

## Data Availability

Data supporting reported results can be found, as indicated at [12].

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
