# Peer review of "Zero-Day Malware Detection and Effective Malware Analysis Using Shapley Ensemble Boosting and Bagging Approach"

_sensors, 2022, doi:10.3390/s22072798_

Round 1
Reviewer 1 Report
- An Introduction, motivation of the paper is not clear.
- There should have been some explanations about the reasons of choosing the features that were used.
- The results need more to analysed and discussed.
Author Response
Comment 1
An Introduction, motivation of the paper is not clear.
Response:
Thank for the concern and critical comments. A summary of introduction and motivation is added in section1, page 4. Immediately after novel contribution to the work listed as bullet points.
Action by Author:
Following text is added in section1, Page4 and highlighted in yellow background.
ML model may detect malware but needs further improvement as false negative misclassfication can leave out detection of malware. The motivation here is to detect them with trends in top features based on Shapley value. The false positive misclassfication causes inconvenience to user and planned to be reduced. These improves the efficiency of ML model. The trend in top features can be used to confirm the correct classification making the ML model robust.
Comment2
There should have been some explanations about the reasons of choosing the features that were used.
Response: Sincere thanks for pointing this out. Some more explanation for including the features from PE header is included. These may help the reader to take information directly than depend on the reference included for same. Some explanation included is highlighted in turquoise color.
Action:
Section 3.1 has added with figure 1 following explanation is added and highlighted in yellow.
The malware authors use custom tools to build executable that results in inconsistent values in the fields of PE header. These inconsistent fields can help distinguish malware and benign software. Hence, the fields from PE header are included as features.
Malware use specific functions in DLL to achieve its objectives. The imports and export function can help distinguish malware and benign software. Hence, these are included as features.
comment3
The results need more to analysed and discussed.
Response: Thanks again for pointing this out. Clarification is added in The table 11 about features in various categories and their contribution and highlighted in yellow.
Action:
- Table 11 is improved to include explanation and highlighted in yellow.
- Section 4.4, page 19 Text is added and highlighted in green
We sincerely thank the reviewer for the positive, encouraging comments on our work. Your valuable inputs have helped us a lot in enhancing the standard of the paper. Thank you very much Professor.

Reviewer 2 Report
This paper introduces an ensemble-based Machine Learning for detecting zero-shot malware detection using shapley score. Overall, the paper addresses a very important problem in cybersecurity. However, the table in this paper is very hard to understand and there are several concerns about the experiment.
- Figure 3 is very unclear. The equation is hard to understand and without a lot of explanation.
- Lacking comparison: I assume all the experiment result is obtained from ensemble-based approach with shapley score. Please clearly state some baselines to evaluate your method. For example, you can report the result of ensemble-approach without using shapely score, or the base approach performance.
- What is the result of using only the 13 features selected via shapley vs. using all features?
- Usually, malware could appear much rare compared with benign programs. The dataset here does not reflect such the real-world scenario.
Author Response
Comment1
This paper introduces an ensemble-based Machine Learning for detecting zero-shot malware detection using shapley score. Overall, the paper addresses a very important problem in cybersecurity. However, the table in this paper is very hard to understand and there are several concerns about the experiment.
Response:
Thanks much for the critical comment. We beleive it will help improve readability and quality of the paper. The table 11 at page 18 is improved to include explanation. It is actually describing the top five features seen in each categories of prediction. The information is complex and has been presented in as easy as possible for understanding.
Action:
Table 11 is improved to include explanation and highlighted in yellow.
Comment2
Figure 3 is very unclear. The equation is hard to understand and without a lot of explanation.
Response:
Thank for the concern and critical comments. The explanation for the algorithm in enhanced in section 3.3, page 9 for more clarity.
Action:
Following explanation is added at section 3.3, Page 10.
- In step1, the base value of shap_explanation object is extracted.
- The logit base value is base start point in waterfall plots in Figure 7.
Comment3
Lacking comparison: I assume all the experiment result is obtained from ensemble-based approach with shapley score. Please clearly state some baselines to evaluate your method. For example, you can report the result of ensemble-approach without using shapely score, or the base approach performance.
Response:
Thanks much for pointing this out. Please allow me to share the view point. The baseline comparison is tabulated in table 12 in page 20 with other works in zero-day malware detection. This is novel work and paper with Shapley value do not exist for malware detection as today.
The SHAP value is used for identifying top features in the ML model and further improving the results of ML model by converting the misclassified samples to right category in True positive and True negative. As all the misclassified sample can be corrected for controlled environment, by the trend in feature, the result reaches 100% accuracy for the 2 datasets. Please note this is not property of ML algorithms but enhancement due to trend in features.
Action:
Following text is added in page 20 and highlighted in green color.
As all the misclassified sample can be corrected by the trends in topmost features, the result of ML model for XGBoost reaches 100% accuracy for the 2 datasets D2 and D3 in controlled environment. Please note this is not property of ML algorithms but enhancement due to trend in topmost features.
Comment4
What is the result of using only the 13 features selected via shapley vs. using all features?
Usually, malware could appear much rare compared with benign programs. The dataset here does not reflect such the real-world scenario.
Response:
Thanks much for the critical concern. All the 2351 features has been used to train the ML models. This is an extensive feature derivation for making efficient ML model. Topmost features are identified and their trends are used for further improvement of ML model to convert misclassification to right classification. The trend of topmost features also can be used for robust detection of true positive and true negative samples. This is one of important insight while predicting unknown samples and removing misclassification. This work tries to compare with other zero-day malware work in table 12, page 22 who use future dataset as unknown malware. We have experimented with more datasets for next month on month until Dec 2017 and results are 96-98% due to 2351 features.
The technique in the work can be applied with other types of malware such as Ransomware, Trojans. It will involve training and detecting trends in topmost features. It is mentioned in conclusion section, page 23, second paragraph.
Spellcheck and Proofreading is done
We sincerely thank the reviewer for the positive, encouraging comments on our work. Your valuable inputs have helped us a lot in enhancing the standard of the paper. Thank you very much Professor.
Reviewer 3 Report
This paper analyzes the results of zero-day malware detection by ML models. It visualizes top features used for the detection and the amount of contribution of each top feature. For this purpose, it calculates SHAP values and converts then into probability scales. Then, it compares the trend for each category of prediction in FP, FN, TN, and TP.
The paper defines malware included in future datasets as zero-day malware. However, this definition is not correct. Future datasets might include zero-day malware, but most of the malware included is already known. This paper claims that the detection accuracy is 98% for future datasets, but actual zero-day malware might not be detected at all. The definition of zero-day malware in [25] is more appropriate: "one which does not match known signature or unknown malware."
As shown in Section 4.9, zero-day malware detection using ML is not new. The previous work proposes new techniques, but this paper just uses existing ML algorithms: XGBoost, LightGBM, Random Forest, and Extratree. Without any effort, this paper accompulishes higher accuracy than [4]. However, malware detection of the next month in this paper is usually easier than that of the next year done by [4]. In addition, it is difficult to compare the results between different datasets.
The paper calculates the SHAP value for each feature and converts it into the probability value. This is a common technique to interpret individual predictions. Section 4.4 shows the trend from the interpretation, but the discussion is shallow. The paper claims that the found trend can be used to make inductive rules for reducing misclassification. However, inductive rules are not shown or evaluated.
Author Response
Comment1
This paper analyzes the results of zero-day malware detection by ML models. It visualizes top features used for the detection and the amount of contribution of each top feature. For this purpose, it calculates SHAP values and converts then into probability scales. Then, it compares the trend for each category of prediction in FP, FN, TN, and TP.
Response: Thanks much for encouraging critical remarks for the work done. These are source of encouragements.
Comment2
The paper defines malware included in future datasets as zero-day malware. However, this definition is not correct. Future datasets might include zero-day malware, but most of the malware included is already known. This paper claims that the detection accuracy is 98% for future datasets, but actual zero-day malware might not be detected at all. The definition of zero-day malware in [25] is more appropriate: "one which does not match known signature or unknown malware."
Response: Thanks much for critical observation. Yes, the definition outlined by you is correct. This work compares with other five zero-day malware work in table 12, page 22. All the referred works use future malware dataset as unknown malware.
Our malware dataset has samples with different SHA hash. Every sample in datasets used in this work has exclusive SHA hash value. It indicate the malware is not known. In addition, these samples are not known to ML model during the training. ML model predicts them efficiently.
We admit the trained model will not detect all types of malware. However, the process outlined in this work can be used with other malware of different variant such as Ramsomeware, Trojan, etc. It require specific training with dataset to build ML model. This has been mentioned in conclusion section, page 23, paragraph 2.
The malware datasets used by other researchers are not in public domain due to privacy issue.
We believe the ML model has not seen the future dataset and able to do predict effectively due to extensive features used. due to All the future malware are having.
Comment3
As shown in Section 4.9, zero-day malware detection using ML is not new. The previous work proposes new techniques, but this paper just uses existing ML algorithms: XGBoost, LightGBM, Random Forest, and Extratree. Without any effort, this paper accompulishes higher accuracy than [4]. However, malware detection of the next month in this paper is usually easier than that of the next year done by [4]. In addition, it is difficult to compare the results between different datasets.
Response: Thanks much for very critical observation and issue faced by researchers for dataset for comparisions. The reason for higher accuracy is wide coverage of features. The features are from PE header and many file related properties. All the features sum upto 2351 for each sample. The way the features are derived and used gives higher accuracy.
The dataset used by many authors are not available due to privacy concern and to compare and share the results.
The novel contributions in this work are to further improve the XGBoost, LightGBM, Random Forest, and Extratree ML model. The topmost features trends has been analyzed for each category of predictions. The trends of topmost feature help overcome misclassfication. These make the ML model more efficient and robust. As all the misclassfied samples can can be correctly classified, the accuracy reach 100% in controlled environment.
Action:
Added following text in section 4.5, highlighted in green.
As all the misclassified sample can be corrected by the trends in topmost features, the result of ML model for XGBoost reaches 100% accuracy for the 2 datasets D2 and D3 in controlled environment. Please note this is not property of ML algorithms but enhancement due to trend in topmost features.
Comment4
The paper calculates the SHAP value for each feature and converts it into the probability value. This is a common technique to interpret individual predictions. Section 4.4 shows the trend from the interpretation, but the discussion is shallow. The paper claims that the found trend can be used to make inductive rules for reducing misclassification. However, inductive rules are not shown or evaluated.
Response:
Thanks much again for the critical observation. It may help improve quality of paper for future readers. The inductive rules are listed in section 4.4, page 19 as bullet points. This portion will be made a new section as improvement and clarity. Discussion will be improved adding explanation.
Action.
1. Separate section is made 4.5 from 4.4 to highlight Inductive rules derived.
2. Text of section 4.4 is replaced with following text. Highlighted in Green.
The water fall plots for false positive sample in Figure 6 and Figure 7 display the topmost features contribute in both positive and negative direction compared to contribution of 2342 features. The observation is same for the topmost features in waterfall plot for false negative sample compared to contribution of 2342 features for false negative sample. However, waterfall plot for true negative sample shows more topmost features contribute in negative directions in same direction as contribution of 2342 features in negative direc-tion. The observation is same for true positive sample. The topmost features contribute in positive direction and in line with contributions of 2342 features. These observations lead to conclude that the waterfall plots in Figure 6 and Figure 7 for false negative, false posi-tive categories are very different from waterfall plots for True positive and True Negative categories. Following two important conclusions are noted.
• The topmost features contribute to different directions than 2342 features for false positive and false negative samples.
• The topmost features contribute to same direction as 2342 features for true positive and true negative samples.
This observation and conclusion can be used for correct classification of misclassified samples. The FP and FN prediction can be identified using waterfall plots. The waterfall plot for the FP and FN categories will show the topmost features for sample will contribute in both positive and negative direction compared to 2342 features. The TP and TN samples can be identified using their waterfall plots. The waterfall plots for TP and TN categories will display the topmost features will contribute in same direction as 2342 features.
We sincerely thank the reviewer for the positive, encouraging comments on our work. Your valuable inputs have helped us a lot in enhancing the standard of the paper. Thank you very much Professor.

Round 2
Reviewer 2 Report
After reading the comments and revised draft. I believe the authors solved some of my concerns. However, there are still two remaining concerns that are not addressed. So I just suggest one round of revision to improve the draft before acceptance.
1. Algorithm 1 is still very hard to understand. For example, what do you mean Logit_base_value = ??????? = 1/1+?−??? ;? Do you repeat the same content twice in each line? It is very hard to understand. Also, the font size changed dramatically in the algorithm. You should use consistent font size and style in the algorithm.
2. Comparison experiment: What I want to see in this paper is an experiment that compared the performance with and without shaply score. For example, you should report the performance of XGBoost which used all features as a baseline to justify your contribution.
Author Response
Concern # 1: Algorithm 1 is still very hard to understand. For example, what do you mean Logit_base_value = ??????? = 1/1+?−??? ;? Do you repeat the same content twice in each line? It is very hard to understand. Also, the font size changed dramatically in the algorithm. You should use consistent font size and style in the algorithm.
Author Response: Thanks much for your critical comments and concern. The logit base value is from the logistic regression logit function (1/1+e(-x)). It is the sigmoid function that transforms value (- ∞ to + ∞) to 0 to 1 on a probability scale. It gives the S curve. {Reference – Chapter 10, Page 248, Introduction to Machine learning, Third edition, Ethem Alpaydin (MIT Press)}. In this algorithm, X= SBV.
The fonts of algorithms have been changed and highlighted in the paper for easy observation. Content is only one time. Space between each line is increased for clarity. We believe it leads to improvement in paper clarity and quality.
Author Action:
- The font is updated in the algorithm.
- Increased space between lines for clarity.
Concern # 2: Comparison experiment: What I want to see in this paper is an experiment that compared the performance with and without a shaply score. For example, you should report the performance of XGBoost which used all features as a baseline to justify your contribution.
Author Response: Thanks much for the observation. Tables 5,6,7, and 8 are baseline results with all the features without shapely.
Figures 4,5,6, and 7 are with Shapley value. Shapley value is applied to find the contribution of features. This identifies features that contributes highest and are termed as top features. These topmost features are compared in table 11 and used to make inductive rules. The inductive rules in section 4.9 are used for correcting the FP and FN samples. As all the FP and FN samples can be put in the correct category the accuracy is 100%. It is mentioned in section 4.9 and the following action.
The contribution is as outlined above.
Author Action:
Addition of the following statement highlighted.
As all the misclassified samples can be corrected by the trends in topmost features, the result of the ML model for XGBoost reaches 100% accuracy for the 2 datasets D2 and D3 in a controlled environment. Please note this is not a property of ML algorithms but enhancement due to the trend in topmost features identified using Shapley values.
Concern # 3: English language and style are fine/minor spell check required
Author Response: Spellcheck and Proofreading were done.
We sincerely thank the reviewer for the positive, encouraging comments on our work. Your valuable inputs have helped us a lot in enhancing the standard of the paper. Thank you very much Professor.

Reviewer 3 Report
The authors claim that each malware dataset has samples with different SHA hash values in the response letter. If so, malware included in the used datasets can be called zero-day malware. However, the paper does not describe how to create such datasets. Please explicitly add that.
Four bullet points listed in Section 4.9 do not seem to be inductive rules. They are just observations in Table 11. Please describe "rules" to apply them to correct misclassification.
The paper claims that XGBoost achieved 100% accuracy after misclassification was corrected. However, the same dataset D2 is used to derive inductive rules and correct misclassification. Please use different datasets, e.g., D2 and D3, between them to show the generality of the obtained inductive rules.
Author Response
Concern # 1: The authors claim that each malware dataset has samples with different SHA hash values in the response letter. If so, malware included in the used datasets can be called zero-day malware. However, the paper does not describe how to create such datasets. Please explicitly add that.
Author Response:: Thanks much for the positive comment that we have used the zero-day malware. The code for the dataset is available at https://github.com/elastic/ember. We have stated in the paper in section 4.1 “Dataset D1, D2, and D3 are extracted from [12] by filtering and processing as per Figure 2. Samples are selected for a dataset based on a specific period. “
We have our code to take any malware sample, extract the features and use the features for prediction using the trained model. Hence, the development in this can be applied to a real-time sample.
We have referred to them in our work rather than describe them again.
Author Action:
Concern # 2: Four bullet points listed in Section 4.9 do not seem to be inductive rules. They are just observations in Table 11. Please describe "rules" to apply them to correct misclassification.
Author Response: Thank you for the concern and critical comments. We have added text for rules for FP, FN, TP, and TN. This improves the quality of the paper and resolves the thought of future reader of the paper.
Author Action: Following inductive rules are added in section 4.9, pages 20-21.
The Inductive rules are derived as follows.
- An unknown real-time sample that has top features as Imp321, C_char1, H33, and Str43. The amount of contribution from 2342 features is opposite to Imp321. It is a FP sample. It is predicted as malware but can be used as benign software.
- An unknown real-time sample that has the top most features as Rx_sec_num and the contribution of 2342 features is opposite to Rx_sec_num features. It is a FN sample. It is predicted as benign but malware. Manual static and dynamic analysis of the sample may be performed.
- An unknown real-time sample that has C_char1 as the topmost feature, and has no contribution of Imp321 is malware as detected. It should have a much higher contribution from 2342
- An unknown real-time sample that has the top most feature as Subsystem9 and contribution from 2342 features in the same direction as Subsystem9 should be a benign sample. It is a verification of the sample and a robust prediction.
Concern # 3: The paper claims that XGBoost achieved 100% accuracy after misclassification was corrected. However, the same dataset D2 is used to derive inductive rules and correct misclassification. Please use different datasets, e.g., D2 and D3, between them to show the generality of the obtained inductive rules.
Author Response: Thanks much for pointing this out. We had used the samples from the D2 dataset in the paper. The picture of the D3 dataset does not add any value. We are sharing a zip file that has bar plots and waterfall plots from the D1 and D2 datasets for ready reference. The caption of Figures 4,5,6, and 7 is updated that the samples are from the D2 dataset. We believe this brings more clarity to the diagram and the paper.
Author Action:
Updated the caption of figures 4,5,6, and 7 as follows.
Figure 6. Waterfall plot of a FP sample (first row left), a FN sample (first row right), a TP sample (second row left), and a TN sample (second row right) from the D2 dataset in shap value.
Shared a ZIP file with images of barplots and waterfall plots from D1 and D2 datasets.
Concern # 4: Moderate English changes required
Author Action: Spellcheck and Proofreading were done.
We sincerely thank the reviewer for the positive, encouraging comments on our work. Your valuable inputs have helped us a lot in enhancing the standard of the paper. Thank you very much, Professor.

Round 3
Reviewer 2 Report
The authors addressed my all concerns.
Author Response
We sincerely thank the reviewer again for the positive, encouraging comments on our work. Your valuable inputs have helped us a lot in enhancing the standard of the paper.
Thank you very much Professor.
Reviewer 3 Report
For the description of creating datasets, it is important that each malware dataset has samples with *different* SHA hash values for zero-day malware. Please add this critical point to Section 4.1.
The last sentence of the 3rd inductive rule seems to be incomplete. Please complete that sentence.
Author Response
Concern # 1: For the description of creating datasets, it is important that each malware dataset has samples with *different* SHA hash values for zero-day malware. Please add this critical point to Section 4.1.
Author Response:: Thanks much for the positive comment we believe it will improve the clarity in the paper. We have added following highlight in section 3.1 (page9) and 4.1.
Author Action:
Section3.1, Page 9.
We use bagging and boosting machine learning algorithm model in sklearn library as shown in the system block diagram in Figure 3. For bagging, Random forest and Extratree forest model and for boosting, gradient boost decision tree based LightGBM and XGboost models are used in this work. Machine learning models using each of the methods such as Random forest, Extratree, LightGBM, and XGBoost are made using dataset D1 that has samples from January 2017. These trained models are used to predict the future zero-day malware and benign samples from February 2017 and March 2017 represented by the D2 dataset and D3 dataset respectively. All the samples in D1, D2, and D3 have different SHA256 values and the D2 and D3 samples are not exposed to training ML models. Few top features in a range of 6-25 features among the 2351 features that help identify the malware and benign samples are visualized using various diagrams such as bar plot and waterfall plot. In addition, these diagrams show the amount of contribution by each top feature in shap value and scale of probability. Any machine learning model learns from a set of given samples, termed as training samples, and can be used for predicting the unseen samples. In this work, the unseen samples are malware and benign samples from the future period such as February 2017 and March 2017 represented by D2 and D3 datasets. Zero-day samples in this work are from future period, have different SHA256 value, not exposed, and unknown to training of ML models. The prediction results are discussed in section IV and are very promising.
Section 4.1 page 11
Dataset D1, D2, and D3 are extracted from [12] by filtering and processing as per Figure 2. Samples are selected for a dataset based on a specific period. The samples in the D1 dataset are from Jan-2017. The counts of unlabeled, malware, and benign samples are described in Table 3. The samples in the D2 dataset are from Feb-2017 and the samples in D3 are from March-2017. All the samples in D2 and D3 have different SHA256 values compared to samples in the D1 dataset, future period, and are not exposed to training the ML models. The counts of samples in unlabeled, malware and benign are in Table 3.
Concern # 2: The last sentence of the 3rd inductive rule seems to be incomplete. Please complete that sentence.
Author Response: Thanks much for your concern and critical comments. We have updated the 3rd inductive rule.
Author Action:
Inductive Rule 3 is updated as follows.
An unknown real-time sample that has C_char1 as the topmost feature, no contribution of Imp321 feature, and much higher contribution from the remaining 2342 features is malware. It is a verification of the sample and a robust prediction.
Concern # 3: Moderate English changes required
Author Action: Spellcheck and Proofreading done.
We sincerely thank the reviewer for the positive, encouraging comments on our work. Your valuable inputs have helped us a lot in enhancing the standard of the paper. Thank you very much Professor.
